# Optimization of the Reduction of Shrinkage and Warpage for Plastic Parts in the Injection Molding Process by Extended Adaptive Weighted Summation Method

**DOI:** 10.3390/polym14235133

**Published:** 2022-11-25

**Authors:** Guillermo Hiyane-Nashiro, Maricruz Hernández-Hernández, José Rojas-García, Juvenal Rodriguez-Resendiz, José Manuel Álvarez-Alvarado

**Affiliations:** 1CIATEQ A. C. Plasticos y Materiales Avanzados, Av. Del Retablo 150, Querétaro 76150, Mexico; 2Facultad de Ingeniería, Universidad Autónoma de Querétaro, Querétaro 76010, Mexico; 3CONACYT, Corporación Mexicana de Investigación en Materiales, Saltillo 25290, Mexico

**Keywords:** injection molding manufacturing, genetic algorithm, gray relational analysis, industrial design for injection molding, Moldflow simulation

## Abstract

The consumer market has changed drastically in recent times. Consumers are becoming more demanding, and many companies are competing to be market leaders. Therefore, companies must reduce rejects and minimize their operating costs. One problem that arises in producing plastic parts is controlling deformation, mainly in the form of shrinkage due to the material and warpage associated with the geometry of the parts. This work presents a novel extended adaptive weighted sum method (EAAWSM: Extended Adaptive Weighted Summation Method) integrated into a Pareto front model. The performance of this model is evaluated against three other conventional optimization methods—Taguchi–Gray (TG), Technique for Order of Preference by Similarity to Ideal Solution (TOPSIS), and Model Optimization by Genetic Algorithm (MOGA)—and compared with EAAWSM. Two response variables and three input factors are considered to be analyzed: material melting temperature, mold temperature, and filling time. Subsequently, the performance is compared and its behavior observed using Moldflow^®^ simulation. The results show that with the EAAWSM method, the shrinkage is 15.75% and the warpage is 3.847 mm, regarding the manufacturing process parameters of a plastic part. This proposed deterministic model is easy to use to optimize two or more output variables, and its results are straightforward and reliable.

## 1. Introduction

The use of injection molding has been growing since the 20th century due to its repeatability and capacity for high-quality mass production and precision castings with various designs and complex geometries. Researchers have studied the process: the raw material, injection equipment, the tooling required for molding, and their interactions [1,2,3,4].

For this, it is necessary to analyze the effect of the deformation and warping of plastic parts in reference to the mechanics of materials and their mechanical characteristics. There are some related works in the literature about the importance of studying the materials and their mechanical characteristics. In [5], the authors developed an investigation on cellulose nanocrystals and the mechanical properties of polyester resins. In [6], the authors optimized the electrophoretic deposition process parameters of polyaniline film. In [7], the authors developed research on the effect of the inclusion of cellulose nanocrystals on the mechanical properties of polyester resins.

Plastic warping is one of the defects that has the most impact economically, and the efficiency results in the production of parts. These defects include shrinkage due to the nature of the material (measured in percentages) and casting warpage (measured in mm of displacement). Different optimization techniques have been used to reduce these defects.

Uddin et al. [8] used shrinkage and warpage reduction optimization using the Taguchi method and Moldflow^®^. Zhao N et al. [9] applied four comparison methods (artificial neural networks, genetic algorithm (GA), response surface methodology, and Kriging model) to optimize process parameters and minimize shrinkage and warpage. They considered five process factors (melting temperature, mold temperature, cooling time, packing time, and injection pressure). In [10], the authors proposed a grey relational comprehensive evaluation model by combining the Technique for Order of Preference by Similarity to Ideal Solution (TOPSIS) with grey relational analysis. This technique achieved a 43.33% reduction in surface marks caused by shrinkage deformation by reducing this deformation by 14.6%, although it did not eliminate warping deformation but maintained it.

Mehat N. et al. [11] studied the effects of processing parameters on a plastic gear and their effects on shrinkage and the mechanical properties of the gear. These models consider using at least eight process factors, which results in an excessive number of experiments, as a product of combinations between factors and levels, which could delay implementation time in a real case and lead to excessive expenditure of resources [12,13,14]. Many authors mention that the factors that affect the deformation of a plastic part are the melting temperature of the material, the mold temperature, and the filling time. However, depending on the complexity of the design, the dimension of the part, and the characteristics of the material, other factors are added, which make the optimization more complex, nonlinear, and unstable over time [15].

Likewise, many works demonstrate the use of more conventional optimization methods for particular cases. Due to this, a mathematical model with fewer factors can save time and resource consumption, giving a deterministic solution to obtain a more reliable and easy to implement product in the industry.

The main contribution and objective of this paper is to present the development of a novel mathematical model called Extended Adaptive Weighted Sum (EAAWSM), which is derived from the Adaptive Weighted Sum model of the conventional Pareto front, and to apply the optimization techniques of the parameters in an injection process. Three control variables selected according to the Taguchi method are considered (material melting temperature, mold temperature, and filling time), whose impact affects the response variables (material shrinkage and part warpage), resulting in the reduction of rejects, which implies a considerable reduction in time and use of resources, thus improving the quality according to the end-expectations of the user. In order to measure the performance of the model, we considered two popular stochastic methods, Taguchi–Gray and TOPSIS, and an evolutive method, multi-objective genetic algorithm (MOGA), in order to compare the performance in the analysis of a plastic part.

## 2. Theoretical Considerations on Injection Molding Process Optimization

This section presents the optimization method and its application given by different authors, referring to the plastic injection molding process. Likewise, the Weighted Sum Method (WSM) model is presented, and three other conventional methods are compared to evaluate their performances.

### 2.1. Optimization Methods in Injection Molding Processes

Optimization methods have evolved over the years. This development has moved from the trial and error method through the development of analytical calculations to the use of statistical models and the use of more advanced techniques such as second-order mathematical models and the use of artificial intelligence [16,17,18,19]. These models consider using existing plant equipment or specialized Engineering Add Computer (CAE) software such as Moldflow^®^ [20]. Multi-objective optimization can not only be used to address these competing problems in injection molding [21,22] but can also be used in other complex applications for novel design. In the innovative hybrid process that has recently been developed to manufacture metal–polymer composites, multi-objective optimization can help to optimize the quality of the polymer casting while still meeting the requirements of the metal part [23,24]. In the fabrication of a hybrid material structure by injection molding polypropylene (PP) with high ductility into a robust thermosetting CF/Epoxy sheet, multi-objective optimization can be used to optimize the shape and bondability of the product.

#### Front-Pareto Optimization Method and the EAAWSM Model

Researchers have developed many mathematical optimization models. Rajesh Kumar et al. (2015) [25] present these concepts of mathematical modeling in their basic form and in a cursory manner. The genetic algorithm is derived from the basic form, whose solution is not mathematical but based on the concept of survival. Bejarano Lilian et al. [26] mention the different algorithms derived from the genetic algorithm. Among the most important are the MOGA and NSGA algorithms.

On the other hand, we have the group whose solution is a deterministic or mathematical solution of the model. Kalyanmoy Deb et al. (2011) [27] presented a model based on the Pareto Optimal Solution (POS), Weighted Sum Method (WSM), ϵ-Contraint Method (CM-ϵ), and Weighted Metric Method (WMM). This group of methods is widely used in complex processes, mainly in the chemical and composite materials industry.

This paper presents a modification of the WSM presented by [28] by considering the restriction of the variables in order to obtain an optimal point and reduce the shrinkage and warpage in the plastic injection process. Regarding the plastic injection molding process, different works use Pareto optimization to optimize the parameters of the injection molding process to optimize the quality of the parts in order to reduce manufacturing costs [29,30,31].

### 2.2. Other Optimization Methods to Be Studied

Three other conventional optimization methods are compared with the EAAWSM method. The Taguchi–Gray method, the TOPSIS method, and the Optimization of Genetic Algorithm (MOGA) method are detailed in the following chapters.

#### 2.2.1. Taguchi–Gray Method

The Taguchi method is a process optimization method that was widely used in the 20th century. It was designed to be robust and optimize operations under changing environmental conditions (noise signals). One of the objectives of the Taguchi method is to reduce the number of tests required, thus improving test efficiency. It was applied to optimize the shrinkage and warpage reduction of plastic processes and then has been compared with the application of finite elements and Moldflow^®^ [32,33,34,35,36]. Gray Relation Analysis (GRA) constitutes a tool with an approach to solving multi-objective optimization problems. The characteristic of this tool is that the Taguchi method is only oriented to optimize a single output variable, while the GRA, which uses multiple factors and multiple variables and has complex interactions, allows us to optimize several variables at the same time [36,37]. Its latest applications and contributions are in the medical and aerospace areas [38].

#### 2.2.2. TOPSIS Optimization Method

TOPSIS (Technique for Order of Preference by Similarity to Ideal Solution) is a multi-variable optimization tool that is widely used today for prioritization in a complex and sophisticated environment [10]. The method enables ranking alternatives in decision-making problems with opposing and contradicting criteria and comparing the distance of each alternative [39]. The best option should have the smallest distance to the positive ideal and the most significant distance to the negative ideal. The purpose of decision-making is to find the most desirable alternatives from a discrete set of feasible options concerning a finite set of attributes. Its use is applied to various areas such as society, economics, the military, management, to name a few. However, its use has spread in process optimization, mainly in the reduction of the shrinkage and warpage of plastic parts [40].

#### 2.2.3. Multi-Variable Genetic Algorithm (MOGA)

Genetic algorithms (GA) are one of the multi-objective optimization methods (MOO) used today. GAs have developed and advanced intensively in recent years, but the fundamental principles remain the same. As Goldberg pointed out, the main principles that make them different from classical methods are as follows: (1) GAs work with a set of points (population) instead of a single one; (2) GAs deal with objective functions directly, and there is no need to derive the functions to find the optimal value; and (3) GA operators have a probabilistic nature, in contrast to the deterministic approaches used in all classical methods. That is why their solution closely resembles the Pareto front solution. It is applied in almost all fields: industry, commerce, education, and others. In chemical industries, it is applied to optimize processes. Zapf F. et al. (2022) [41] proposed the implementation of the multi-objective optimization of the naphtha catalytic reformer. In [42,43], the authors carried out similar multi-objective cases for a non-conventional naphtha catalytic reformer where aromatics, hydrogen, and aniline production were maximized. The authors of [44] investigated the effectiveness of a Genetic Algorithm applied in solving a Logistics Engineering problem. The next chapter applies MOGA to optimize a plastic injection process.

## 3. Methodology

At the beginning of these optimization methods, it is necessary to make a baseline, where the initial conditions of the process are presented and form the base reference for the changes to be executed. Subsequently, the Pareto front method is presented in detail. A flow diagram of the model is presented, and each step of the model is explained. Subsequently, the three comparative optimization methods are shown—(1) Taguchi–Gray (GRA), (2) multi-criteria decision-making techniques (TOPSIS), and (3) the Genetic Algorithm (MOGA)—and their efficiencies are compared with the presented model.

### 3.1. Materials and Equipment

The material used to perform the analysis was the thermoplastic polymer polypropylene (PP 1032). The mechanical features of the material were considered from Exxon Chemical (United States) for the simulation. The characteristics of the material are shown in Table 1.

In terms of injection equipment, a Haitian 120 TN machine was used. The equipment characteristics are shown in Table 2. The injection mold used in this research is a commercial brand with a P-20 steel core and cavity inserts.

### 3.2. Plastic Part

In order to perform the experiment, a refrigerator handle was taken as a test bench. The databasewas imported into the IGES version of Solidwork, which allowed it to be used in the Moldflow^®^ simulator. The casting is shown in Figure 1.

### 3.3. Experimental Development

It is necessary to define a baseline to establish the current conditions of a process or where the problem is identified.

#### Baseline

The initial process conditions are shown in Table 3. The significant factors are as follows: material temperature (*X*${}_{1}$), mold temperature (*X*${}_{2}$), and part filling time (*X*${}_{3}$). The other factors are not significant because, statistically, they had a *p*-value ≥1 [45].

### 3.4. EAAWSM Experiment

Figure 2 shows the flow chart of the Extended Weighted Sum Method (EAAWSM), which describes the Extended and Modified Pareto Front Method to reduce and select the process factors that have the most significant impact on the shrinkage and warpage of the part to be studied.

The initial experiment is presented with eight factors and two output variables. The factors were as follows: material melting temperature (*X*${}_{1}$), mold temperature (*X*${}_{2}$), plastic part filling time (*X*${}_{3}$), injection pressure (*X*${}_{4}$), packing pressure (*X*${}_{5}$), injection speed (*X*${}_{6}$), packing time (*X*${}_{7}$), and part cooling time (*X*${}_{8}$). The output variables were as follows: material shrinkage (*Y*${}_{1}$) and warpage (*Y*${}_{2}$), measured in percentages (%) and millimeters (mm), respectively. The initial number of experiments was 38 runs, i.e., 6561 experiments. This would demand many resources in terms of time, material, equipment, and energy. The number of experimental runs through the Taguchi method consisted of an L27 Orthogonal Array, whose purpose is the reduction of factors. Subsequently, an ANOVA was performed. It was determined that only three factors were significant (with *p*-values ≥0.05). These were *X*${}_{1}$, *X*${}_{2}$, and *X*${}_{3}$. Subsequently, noise signals (N/S) were calculated for each output variable *Y*${}_{1}$ and *Y*${}_{1}$. The results are presented in Appendix A.

Step 1: Definition of the experimental run (Taguchi): A DOE–Taguchi is defined, with 27 experiments through an Orthogonal Array L27, where the runs, their sequence, and the levels to be combined have to be defined. The purpose is to obtain the output variables of shrinkage and warping resultsStep 2: Simulation of the runs through Moldflow^®^ to obtain results. Through this CAE tool, the injection process of the part to be analyzed can be simulated, and the results required for each process run can be obtained. According to the method, there are 33 runs described in the design of experiments, taking as control variables the material temperature (*X*${}_{1}$), mold temperature (*X*${}_{2}$), and filling time (*X*${}_{3}$).Step 3: Include quadratic variables and double interactions. The Taguchi method does not consider quadratic variables and double interactions. It is appended to the model in order to increase the reliability of the model. In this step, the output variables of the process are obtained independently—shrinkage and warpage—according to Equations (Equation 1) and (Equation 2):


(1)
$$\begin{array}{c} Y_{1}=\beta_{01}+\beta_{11}X_{1}+\beta_{21}X_{2}+\beta_{31}X_{3}+\beta_{111}X_{1}^{2}+\beta_{221}X_{2}^{2}\\ +\beta_{331}X_{3}^{3}+\beta_{121}X_{1}X_{2}+\beta_{131}X_{1}X_{3}+\beta_{231}X_{2}X_{3} \end{array}$$



(2)
$$\begin{array}{c} Y_{2}=\beta_{02}+\beta_{12}X_{1}+\beta_{22}X_{2}+\beta_{32}X_{3}+\beta_{112}X_{1}^{2}+\beta_{222}X_{2}^{2}\\ +\beta_{332}X_{3}^{3}+\beta_{122}X_{1}X_{2}+\beta_{132}X_{1}X_{3}+\beta_{232}X_{2}X_{3} \end{array}$$


Step 4: Define the optimal function and the scale parameters $\lambda_{1}$ and $\lambda_{2}$: Functions $f_{1}$ and $f_{2}$ are defined and associated with the scaling parameters by Equations (Equation 3) and (Equation 4):


(3)
$$f_{1}=Y_{1}\lambda_{1}$$



(4)
$$f_{2}=Y_{2}\lambda_{2}$$


and according to Equation Equation 5:(5)$$\lambda_{1}+\lambda_{2}=1$$

We define the optimum function $f_{op}$ given by
(6)$$f_{op}=f_{1}+f_{2}=Y_{1}\lambda_{1}+Y_{2}\lambda_{2}$$

Step 5: Then, we can calculate the optimal values of variables *X*${}_{1}^{*}$, *X*${}_{2}^{*}$, and *X*${}_{3}^{*}$ of the input variables *X*${}_{1}$, *X*${}_{2}$, and *X*${}_{3}$, respectively. For this, the optimal function $f_{op}$ is derived and equals 0. Having three variables will form three simultaneous linear equations:


(7)
$$X_{1}^{*}=\frac{\partial f_{op}}{\partial X_{1}}$$



(8)
$$X_{2}^{*}=\frac{\partial f_{op}}{\partial X_{2}}$$



(9)
$$X_{3}^{*}=\frac{\partial f_{op}}{\partial X_{3}}$$


The following simultaneous equations are formed:(10)$$C_{1}=a_{11}X_{1}^{*}+a_{12}X_{2}^{*}+a_{13}X_{1}^{*}$$
(11)$$C_{2}=a_{21}X_{1}^{*}+a_{22}X_{2}^{*}+a_{23}X_{1}^{*}$$
(12)$$C_{1}=a_{31}X_{1}^{*}+a_{32}X_{2}^{*}+a_{33}X_{1}^{*}$$

Then, the values of *X*${}_{1}^{*}$, *X*${}_{2}^{*}$, and *X*${}_{3}^{*}$ are calculated.

Step 6: Place variables *X*${}_{1}^{*}$, *X*${}_{2}^{*}$, and *X*${}_{3}^{*}$ under constraints. These optimal values must be constrained according to the limit ranges of the restriction of variables specified in the defined levels:


(13)
$$Ll_{1}\leq X_{1}^{*}\leq Ls_{1}\;\mathrm{if}\;X_{1}^{*}\neq \left[Ll_{1},Ls_{1}\right]\;,\;\mathrm{therefore}:\;X_{1}^{*}=0$$



(14)
$$Ll_{2}\leq X_{2}^{*}\leq Ls_{2}\;\mathrm{if}\;X_{2}^{*}\neq \left[Ll_{2},Ls_{2}\right]\;,\;\mathrm{therefore}:\;X_{2}^{*}=0$$



(15)
$$Ll_{3}\leq X_{3}^{*}\leq Ls_{3}\;\mathrm{if}\;X_{3}^{*}\neq \left[Ll_{3},Ls_{3}\right]\;,\;\mathrm{therefore}:\;X_{3}^{*}=0$$


Step 7: Define the graph: We plot the *Y*${}_{op}$ versus λ values, as shown in Figure 3, where the sets of points other than zeros are specified as feasible solutions. The optimal solution is the highest value.

Step 8: Define the values *X*${}_{1}^{*}$, *X*${}_{2}^{*}$, and *X*${}_{3}^{*}$: In this step, the statistical equations obtained in step 3 must be defined, and the optimal values in the equations must be replaced to obtain the required minimum shrinkage and warping values.

## 4. Experimental Results

This section shows the results of the proposed EAAWSM method and a comparison to evaluate its performance against three other conventional methods.

### 4.1. Application of the EAAWSM Method

Table 4 shows the values that the quadratic effects must achieve to improve the model. Here, the Minitab 15 software was used to obtain the parameters.

Next, Equation (Equation 16) shows the shrinkage result:(16)$$Y_{1}=65.8-0.373X_{1}+0.0523X_{2}-6.42X_{3}+0.000821X_{1}^{2}-0.000517X_{2}^{2}+1.27X_{3}^{2}$$
and Equation (Equation 17) shows the warpage result:(17)$$\begin{array}{c} Y_{2}=-0.57+0.0229X_{1}-0.0164X_{2}+1.62X_{3}-\\ 0.000043X_{1}^{2}+0.000096X_{2}^{2}-0.331X_{3}^{2} \end{array}$$

Equation (Equation 18) is the optimal Y optimum function, defined by
(18)$$\begin{array}{c} Y_{op}=\left(65.8-0.373X_{1}+0.0523X_{2}-6.42X_{3}+0.000821X_{1}^{2}-0.000517X_{2}^{2}+1.27X_{3}^{2}\right)\lambda_{1}+\\ \left(Y_{2}=-0.57+0.0229X_{1}-0.0164X_{2}+1.62X_{3}-0.000043X_{1}^{2}+0.000096X_{2}^{2}-0.331X_{3}^{2}\right)\lambda_{2} \end{array}$$

The values of λ were simulated, where they varied from 1 to 100. Figure 4 shows the result of the EAAWSM method graph obtained in Program R. The highest value was taken. The result was *X*${}_{1}=227.1685$${}^{\circ }$C, *X*${}_{2}=50.83867$${}^{\circ }$C, and *X*${}_{3}=2.522186$ s. The optimum values of shrinkage and warpage were *Y*${}_{1}=15.85$% and *Y*${}_{2}=3.15$ mm, respectively.

The CAE simulation was performed considering the values *X*${}_{1}$: material temperature = 227.1686 ${}^{\circ }$C, *X*${}_{2}$: mold temperature = 50.8367 ${}^{\circ }$C, and *X*${}_{3}$: filling time of 2.52186 s. The results of the Moldflow^®^ were a shrinkage of 16.02% and a warpage of 3.146 mm, shown in Figure 5.

### 4.2. Results of the Comparative Methods

The analysis of variance (ANOVA) allows us to calculate the optimal factors and levels in the Taguchi–Gray and TOPSIS methods. The MOGA method considered double factors and interactions between two factors in the statistical equation.

The calculations obtained for the Taguchi–Gray method, taking the work of [46] as a reference, are detailed in Appendix B, where their calculations for each step are presented. Appendix C presents the ANOVA analysis, where the optimums are found with the melt temperature at 280 ${}^{\circ }$C, mold temperature at 20 ${}^{\circ }$C, and filling time at 3 s. The results for shrinkage and warpage were 18.63% and 4.04 mm, respectively.

For the calculation of the TOPSIS method, the work of [47] was taken as a reference. The calculations of each step are shown in Appendix D. An ANOVA analysis is presented in Appendix E. In that table, the optimums are found with the material melting temperature at 240 ${}^{\circ }$C, the mold temperature at 60 ${}^{\circ }$C, and the filling time at 3 s. Shrinkage values were 17.06%, and the warpage was 3.68 mm. For the MOGA method, the data input of the experiment considering the quadratic and double factors is presented in Appendix F. For the calculations of the MOGA development, Matlab was considered. Appendix G displays the graph obtained from the optimum results: the melting temperature of the material was 241.4106 ${}^{\circ }$C, the mold temperature was 20.0008 ${}^{\circ }$C, and the filling time was 2.9605 s. The shrinkage results were 15.8391%, and the warpage was 4.1409 mm.

### 4.3. Comparison of Results

In Table 5, the efficiencies of each method are grouped according to the results obtained. EAAWSM was observed as the most efficient among the other three methods.

Then, the results obtained in the simulation were tested through Moldflow^®^ and are shown in Table 6. The EAAWSM method was also optimal in both outputs. It should be noted that the results obtained in Moldflow^®^ in Taguchi–Gray and TOPSIS were the same as they took the same levels.

According with the performance of the mathematical model and Moldflow, it is shown that we obtained the same results (Table 5 and Table 6). Figure 6a shows similar behavior in the Taguchi–Gray method, as well as in Figure 6b in the TOPSIS method and Figure 6c with the MOGA method. The Moldflow represents the confirmation of the physical model. It is observed that the EAAWSM model shows the best result compared to the other three conventional methods: in terms of shrinkage, it was 16.02% versus 18.90% obtained in Taguchi–Gray and TOPSIS and 16.73% compared to MOGA. Similarly, we observed values of 3.15 mm versus 3.17 mm with Taguchi–Gray and TOPSIS and 4.12 mm with MOGA for warpage.

## 5. Discussion

The conventional WSM method proposes the optimization in a single point, according to the scale parameters and without restrictions. However, the proposed EAAWSM method varies the values of the scale parameters by obtaining several points and making a value equal to 0 to those that are outside the range, and those that comply limit them to a range of solutions.

The proposed model benefits from easy calibration and integration compared to the reported methods in the state of the art, as it is based on the utilization of deterministic operations (mathematical) and not stochastic operations (probabilistic) used in Genetic Algorithms and their derivatives.

This method contributes (1) a method with easy application based on basic concepts of optimization models; (2) a reduction of variables, which allows for easy adjustments—at the beginning of the experiment, there were eight factors, which were then reduced to three significant factors through an ANOVA analysis; (3) the reduction of experiments, which brings a significant reduction in computational and executional time, allowing for quicker decision making with fewer resources (materials, use of equipment, indirect materials or energy)—the number of experiments was reduced from 6561 to 27 runs; (4) a model that is mathematical, not probabilistic, which means it is free of uncertainty; (5) a method that quickly identifies the expectations; and (6) a reduction of the risks associated with a real experiment or a field experiment. Table 7 shows a comparative summary of the main inputs of this work versus the conventional models in the literature. It also presents the three control variables and two input variables from this model versus the others, which show up to five variables.

## 6. Conclusions

This article presented a mathematical model for optimization, called EAAWSM, derived from the Pareto front, that optimizes the reduction of the warpage of the two output variables—the shrinkage and warpage—based on the three input variables. It also compared the performance of the EAAWSM method versus that of the three other methods—Taguchi–Gray, TOPSIS and MOGA—which used the three control variables and the two output variables as inputs. The results obtained by EAAWSM were a shrinkage of 15.8% and warpage of 3.68 mm compared to those obtained by the other methods: 18.9% and 4.30 mm (Baseline), 16.9% and 3.9 mm (Taguchi–Gray), 16.82 and 3.68 mm (TOPSIS), and 15.8% and 4.8 mm (MOGA). However, EAAWSM model is limited by every independent variable that is constituted of a series of equations that limit their values. In order to highlight the performance of the proposed model, the main outcomes can be presented as follows

The three factors that directly affect shrinkage and warpage of a casting are the material melting temperature (*X*${}_{1}$), mold temperature (*X*${}_{2}$), and filling time (*X*${}_{3}$).The reduction in warpage of a plastic part is specifically particular depending on the complexity of its geometry, its size, and the type of material injected.This method has better performance as it is deterministic, compared to other methods, such as the Genetic Algorithm, in which the solution is based on the survival of the species.

A proposal for future work includes optimizing the reduction in deformation by changing the injection points and observing its effects in warpage. The proposal also includes using machine learning techniques or other artificial intelligence tools considering the geometry of the piece, the size, and type of material to standardize the search for factors in processes.

## Figures and Tables

**Figure 1 polymers-14-05133-f001:**
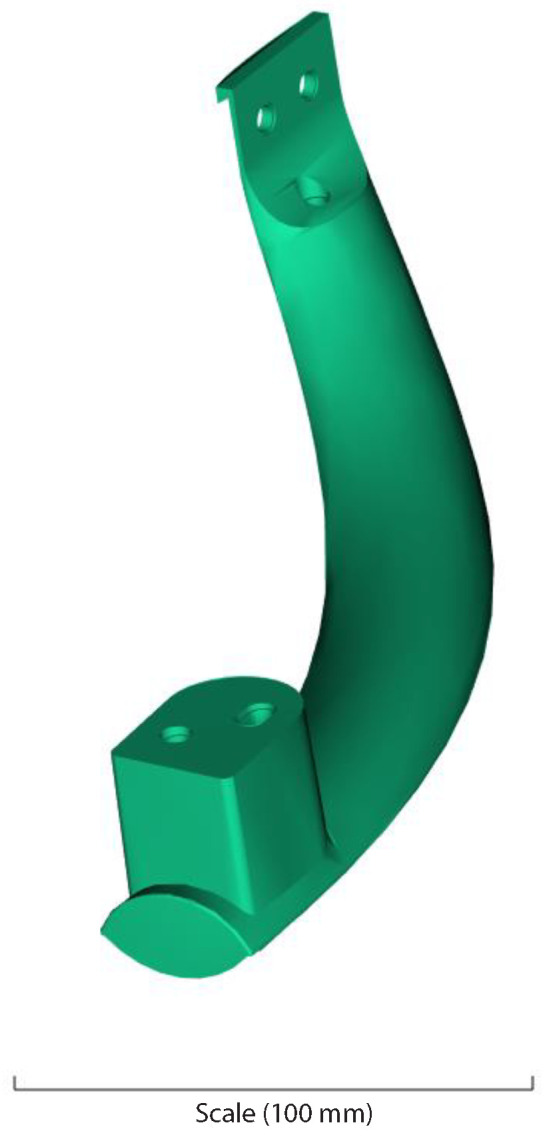
Plastic part.

**Figure 2 polymers-14-05133-f002:**
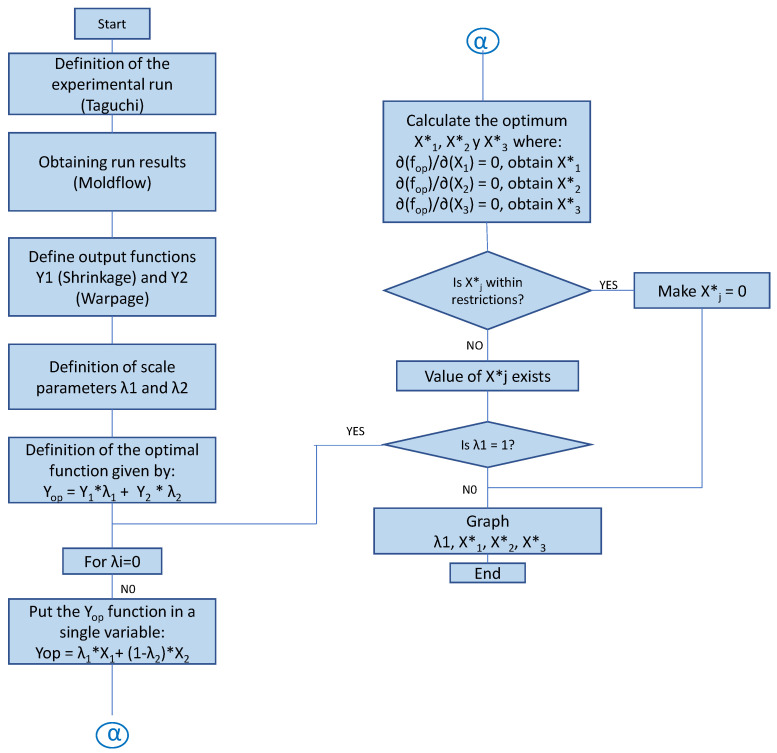
Flowchart for the EAAWSM method.

**Figure 3 polymers-14-05133-f003:**
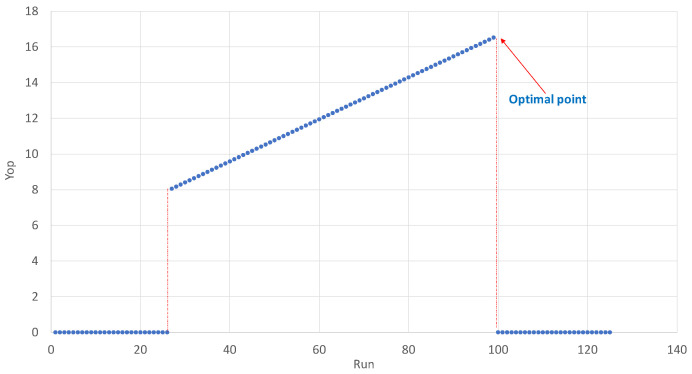
Getting the optimal value.

**Figure 4 polymers-14-05133-f004:**
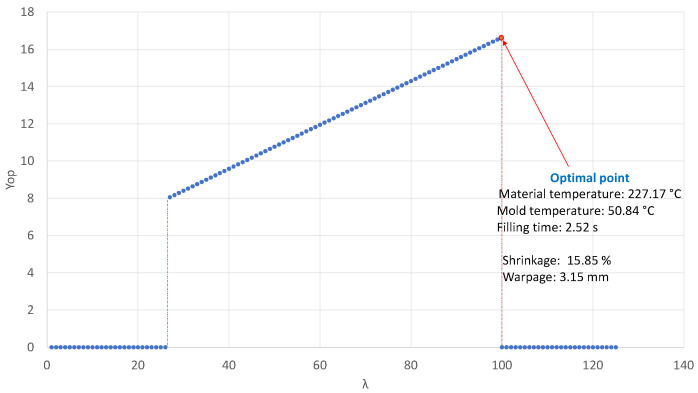
Result of the EAAWSM experiment.

**Figure 5 polymers-14-05133-f005:**
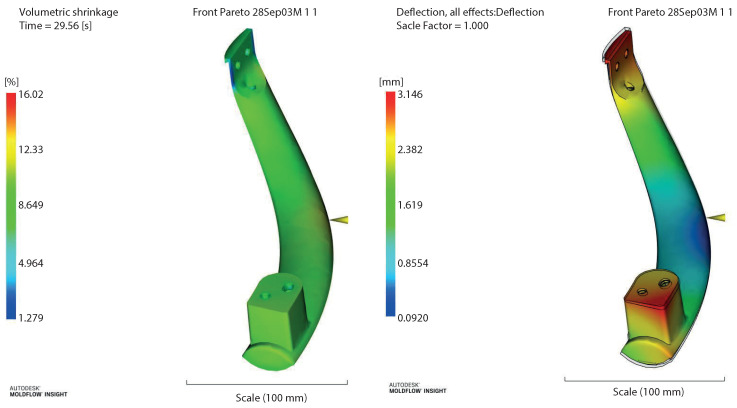
Moldflow^®^ results in the EAAWSM method.

**Figure 6 polymers-14-05133-f006:**
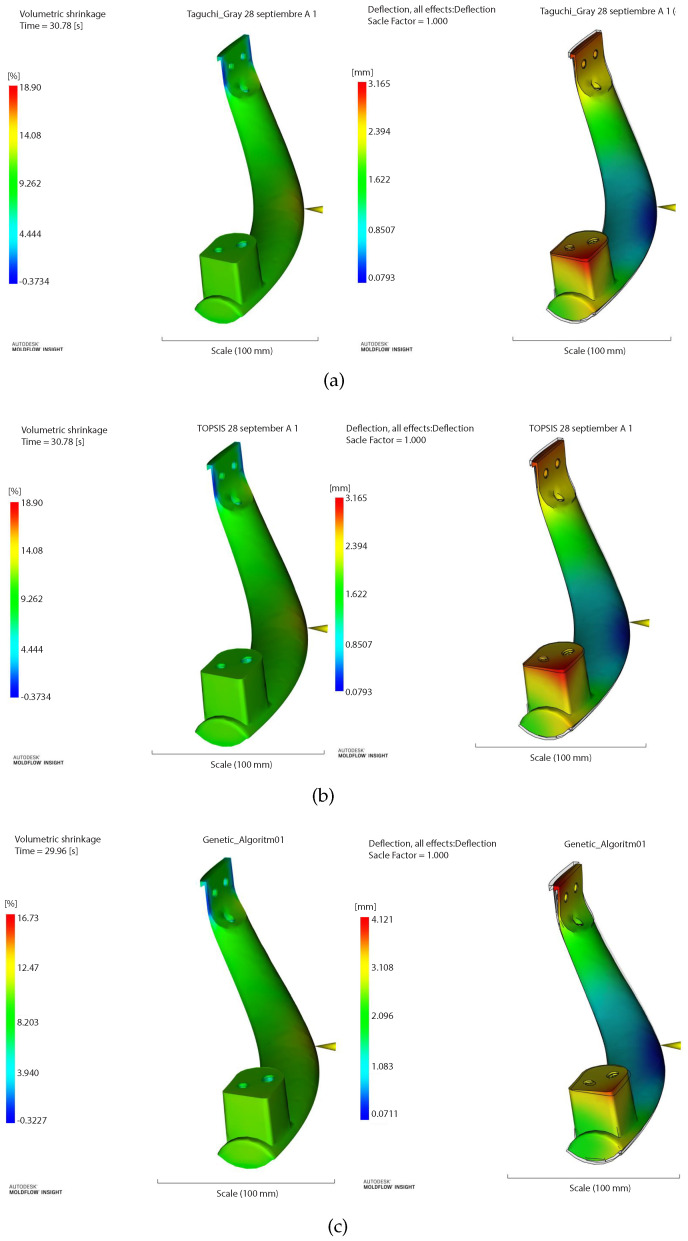
Simulation results of Moldflow of (**a**) Taguchi–Gray, (**b**) TOPSIS, and (**c**) MOGA.

**Table 1 polymers-14-05133-t001:** PP 1032 Material features.

Characteristics	Value	Units
Material: POLYPROPYLENE	-	-
Commercial name: PP-1032	-	-
Supplier: EXXON CHEMICAL USA	-	-
Material melting temperature	240–280	${}^{\circ }$C
Mold melting temperature	20–260	${}^{\circ }$C
Density (visco-elastic state)	0.75967	g/cm${}^{3}$
Density (solid state)	0.92689	g/cm${}^{3}$
Elastic module	1340	Mpa
Poisson ratio	0.392	-
Shear stress modulus	482.3	MPa

**Table 2 polymers-14-05133-t002:** Equipment specifications.

General Description	Value	Units
Closing force (mechanical)	120	TN
Injection capacity (PS reference)	200	g
Stroke diameter	40	mm
Weight per dose	195	g/PS
Maximum opening	360	mm
Minimum ejection opening	120	mm
Plate size	615 × 615	mm

**Table 3 polymers-14-05133-t003:** Baseline reference parameters.

Factors	Values	Units
Input variables		
Melting temperature ($X_{1}$)	280	${}^{\circ }$C
Mold temperature ($X_{2}$)	20	${}^{\circ }$C
Filling time ($X_{3}$)	3	s
Output variables		
Shringkage ($Y_{1}$)	280	${}^{\circ }$C
Warpage ($Y_{2}$)	20	${}^{\circ }$C

**Table 4 polymers-14-05133-t004:** *Y*${}_{1}$ and *Y*${}_{2}$ results from the Minitab.

Regression Results for *Y*${}_{1}$					Regression Results for *Y*${}_{2}$				
Predictor	Coefficient	Coef of EE	T	P	Predictor	Coefficient	Coef of EE	T	P
Constant	65.81	12.54	5.25	0	Constant	−0.57	1.33	−0.43	0.673
$X_{1}$	−0.37292	0.09576	−3.89	0.001	$X_{1}$	0.02286	0.01015	2.25	0.036
$X_{2}$	0.05233	0.01488	3.52	0.002	$X_{2}$	−0.016383	0.001578	−10.38	0
$X_{3}$	−6.423	1.475	−4.35	0	$X_{3}$	1.618	0.1564	10.34	0
$X_{1}^{2}$	0.0008208	0.0001841	4.46	0	$X_{1}^{2}$	−0.00004292	0.00001952	−2.2	0.04
$X_{2}^{2}$	−0.0005167	0.0001841	−2.81	0.011	$X_{2}^{2}$	0.00009583	0.00001952	4.91	0
$X_{3}^{2}$	1.2733	0.2946	4.32	0	$X_{3}^{2}$	−0.33067	0.03123	−10.59	0
S = 0.180398; R-quad = 97.3%; R-quad (adjusted) = 96.5%					S = 0.0191242; R-quad = 98.8%; R-quad (adjusted) = 98.5%				

**Table 5 polymers-14-05133-t005:** Efficiency comparisons between the EAAWSM vs. Taguchi-Gray, TOPSIS and MOGA methods.

Experimental Design	Input Factors *X*${}_{i}$ *Y*${}_{1}$			Model Result	
	*X* ${}_{1}$	*X* ${}_{2}$	*X* ${}_{3}$	*Y* ${}_{1}$	*Y* ${}_{2}$
Baseline	280 ${}^{\circ }$C	20 ${}^{\circ }$C	3 s	18.90%	4.26 mm
EAAWSM	227.17 ${}^{\circ }$C	50.84 ${}^{\circ }$C	2.52 s	15.85%	3.15 mm
Taguchi-Gray	280 ${}^{\circ }$C	20 ${}^{\circ }$C	3 s	18.63%	4.04 mm
TOPSIS	280 ${}^{\circ }$C	20 ${}^{\circ }$C	3 s	17.06%	3.68 mm
MOGA	241.41 ${}^{\circ }$C	20.008 ${}^{\circ }$C	2.9605 s	15.84%	4.14 mm

**Table 6 polymers-14-05133-t006:** Comparisons in the confirmation of the Moldflow^®^ EAAWSM method vs. Taguchi–Gray, TOPSIS, and MOGA.

Experimental Design	Input Factors *X*${}_{i}$ *Y*${}_{1}$			Moldflow Result	
	*X* ${}_{1}$	*X* ${}_{2}$	*X* ${}_{3}$	*Y* ${}_{1}$	*Y* ${}_{2}$
Baseline	280 ${}^{\circ }$C	20 ${}^{\circ }$C	3 s	-	-
EAAWSM	227.17 ${}^{\circ }$C	50.84 ${}^{\circ }$C	2.52 s	16.02%	3.15 mm
Taguchi–Gray	280 ${}^{\circ }$C	20 ${}^{\circ }$C	3 s	18.90%	3.17 mm
TOPSIS	280 ${}^{\circ }$C	20 ${}^{\circ }$C	3 s	18.90%	3.17 mm
MOGA	241.41 ${}^{\circ }$C	20.008 ${}^{\circ }$C	2.9605 s	16.73%	4.12 mm

**Table 7 polymers-14-05133-t007:** Comparisons of similar works in optimization methods for strain reduction.

Author	Optimization Model	Considered Variables	Optimization Method
Our work	Material shrinkage reduction	Material temperature	Weighted Sum Method
	and casting warpage	Mold emperature	Modified (WSMM)
		Filling time	
[12]		Material temperature	
	Reduction of sink mark	Mold temperature	Genetic Algorithm
	and casting warpage	Filling time	NSGA-II
		Packing pressure	
[48]		Material temperature	
	Reduction of sink mark	Mold temperature	Use of Pareto front through
	and casting warpage	Filling time	the Genetic Algorithm
		Cooling temperature	
		Cooling time	
[14]		Material temperature	
	Reduction of sink mark	Injection pressure	Taguchi conventional
	and casting warpage	Packing pressure	technique
		Packing time	
		Cooling time	
[49]		Material temperature	
	Two output variables:	Mold temperature	Taguchi Ccnventional
	shrinkage and warpage	Filling time	technique for equations and
		Packing time	particle swarm optimization
		Cooling time	

## Data Availability

Not applicable.

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
