# Peer review of "Optimization of the Reduction of Shrinkage and Warpage for Plastic Parts in the Injection Molding Process by Extended Adaptive Weighted Summation Method"

_polymers, 2022, doi:10.3390/polym14235133_

Round 1

Reviewer 1 Report

The authors reported a mathematical study for optimizing the reduction of shrinkage and warpage of plastics parts in the injection molding process called Extended Adaptive Weighted Summation Method (EAAWSM). A benchmark study on three other methods: Taguchi-Gray, TOPSIS and MOGA were presented as well. The paper is clearly written and structured. I recommend acceptance with a minor revision. A few comments are listed:

1. Caption of table 5 needs to be revised.

2. In page 11, line 288, there is a sentence “Table 10 shows a comparative summary….”. It should be Table 7.

3. The caption of table 7 is called Equipment specifications. It is same as the one of table 2. Please change it accordingly.

4. Figure 6 (b) is same as Figure (a). Both are Taguchi-Gary. Please revise Figure (b)

Author Response

The authors reported a mathematical study for optimizing the reduction of shrinkage and warpage of plastics parts in the injection molding process called Extended Adaptive Weighted Summation Method (EAAWSM). A benchmark study on three other methods: Taguchi-Gray, TOPSIS and MOGA were presented as well. The paper is clearly written and structured. I recommend acceptance with a minor revision. A few comments are listed:

Answer.- Thanks for your observations. We are glad to follow all your concerns in order to improve the manuscript.

  1. Caption of table 5 needs to be revised.

Answer.- Thanks for your comment. The caption of table 5 was changed as follows:

Table 5 Efficiency comparisons between the EAAWSM vs Taguchi-Gray, TOPSIS and MOGA methods

  1. In page 11, line 288, there is a sentence “Table 10 shows a comparative summary….”. It should be Table 7.

Answer.- Thanks for your comment. We fixed the number of the table in the main text.

  1. The caption of table 7 is called Equipment specifications. It is same as the one of table 2. Please change it accordingly.

Answer.- Thanks for your observation. The caption of table 7 was corrected and it is now named as follows:

Table 7 Comparisons of similar works in optimization methods for strain reduction.

  1. Figure 6 (b) is the same as Figure (a). Both are Taguchi-Gary. Please revise Figure (b)

Answer.- We appreciate your observation. We placed the correct Figure for the TOPSIS method in Figure 6(b).

Reviewer 2 Report

This manuscript tries to figure out an improved method to calculate the shrinkage and warpage. This work can help to optimize the parameters. I recommend that this paper can be published after revision as indicated.

1)      Line 62, MOGA, the first time appears in the text, the full name is required.

2)       Line 96, Because this paper tries to modify the EAAWSM presented by [27], the results should also be compared with the original EAAWSM or WSM.

3)      Line108, “corroborate with”? collaborate with?  

Author Response

This manuscript tries to figure out an improved method to calculate the shrinkage and warpage. This work can help to optimize the parameters. I recommend that this paper can be published after revision as indicated.

Answer.- Thank you for your recommendation. We are glad to improve the manuscript according to your observations.

1)  Line 62, MOGA, the first time appears in the text, the full name is required.

Answer: Thank you for your comments. Multi-objective genetic algorithm (MOGA) is now defined in the main text.

2)  Line 96, Because this paper tries to modify the EAAWSM presented by [27], the results should also be compared with the original EAAWSM or WSM.

Answer.- Thanks for your comment. The WSM is a mathematical model that proposes optimization at a single point, according to given scale parameters; however, this point is not associated with the restrictions of the variables, which means that the optimum is outside the range. The EAAWSM method varies the values of the scale parameters by obtaining several points and making a value equal to 0 to those that are outside the range and to those that comply it limits them to a range of solutions.

We added the following paragraph to the Discussion section (lines 289-293)

The conventional WSM method proposes the optimization in a single point, according to the scale parameters and without restrictions. However, the proposed EAAWSM method varies the values of the scale parameters by obtaining several points and making a value equal to 0 to those that are outside the range and those that comply limit them to a range of solutions.

3)  Line108, “corroborate with”? collaborate with?

Answer: Thanks for your observation. We improved for a better comprehension as follows:

It was applied to optimize the shrinkage and warpage reduction of plastic processes and then compared with the application of finite elements and Moldflow®

Reviewer 3 Report

The paper presents an interesting approach based on the Optimization of the reduction of shrinkage and warpage for plastic parts in the injection molding process by Extended Adaptive Weighted Summation Method. However, the innovation of the current research work should be further highlighted and emphasized. At the same time, the authors should consider the following comments to greatly improve the quality of the paper.

1. In the abstract, add a final statement that highlights the importance of this research and its possible potentials.

2. The introduction needs to be improved by relating to the mechanics of the studied materials and their mechanical characteristics. The references to be included are: 10.1177/0021998318790093, 10.1016/j.polymertesting.2017.09.009, 10.1016/j.compstruct.2021.114698, 10.1177/0731684417727143, 10.1002/app.46770, 10.1016/j.porgcoat.2022.107015.

3. Why exactly the models presented in this study were the only methods taken into consideration?

4. Instead of writing: "The plastic material is Polypropylene", it should be [Thermoplastic] and use the past tense to define what was used.

5. The line stating: "the material was considered to be from Exxon Chemical", was it considered or purchased from that company already?

6. The conclusion needs to be modified to summarize the research outcomes in short statements with clear observations.

Author Response

The paper presents an interesting approach based on the Optimization of the reduction of shrinkage and warpage for plastic parts in the injection molding process by the Extended Adaptive Weighted Summation Method. However, the innovation of the current research work should be further highlighted and emphasized. At the same time, the authors should consider the following comments to greatly improve the quality of the paper.

Answer.- Thank you for your comments. We will take this into consideration to improve the quality of the manuscript.

  1. In the abstract, add a final statement that highlights the importance of this research and its possible potentials.

Answer: Thanks for your observation. We added the next sentence at the end of the Abstract section to highlight the importance of the study.

The model is deterministic and easy to use to optimize two or more output variables, and its results are straightforward and reliable.

  1. The introduction needs to be improved by relating to the mechanics of the studied materials and their mechanical characteristics. The references to be included are: 10.1177/0021998318790093, 10.1016/j.polymertesting.2017.09.009, 10.1016/j.compstruct.2021.114698, 10.1177/0731684417727143, 10.1002/app.46770, 10.1016/j.porgcoat.2022.107015. 

Answer: Thanks for your recommendation. We added a paragraph about the effect of mechanical properties to highlight the importance of studying the materials and their mechanical characteristics in lines 24-31.

For this, it is necessary to analyze the effect of deformation and warping of plastic parts about the mechanics of materials and their mechanical characteristics. There are some related works in the literature about the importance of studying materials and their mechanical characteristics. In \cite{zaghloul2019fatigue} developed an investigation on cellulose nanocrystals on the mechanical properties of polyester resins. In \cite{fuseini2022investigation} optimized the electrophoretic deposition process parameters of polyaniline film. \cite{shchegolkov2022effect} developed research on the effect of the inclusion of cellulose nanocrystals on the mechanical properties of polyester resins. 

  1. Why exactly the models presented in this study were the only methods taken into consideration?

Answer: Thank you for your observation. Two stochastic models such as Taguchi-Gray and TOPSIS were taken into account because their relevance in the optimization of plastic injection processes according to the literature, and MOGA was considered because it is a different and recent method based on evolutionary algorithms, making it different from the mathematical ones. These models served as a reference to compare the performance of the model that we propose. We place the following sentence at the end of the Introduction section (lines 70-73):

In order to measure the performance of the model, it was compared with two popular stochastic methods: Taguchi-Gray and TOPSIS, and an evolutive method: the objective genetic algorithm (MOGA).

  1. Instead of writing: "The plastic material is Polypropylene", it should be [Thermoplastic] and use the past tense to define what was used.

Answer.- Thanks for your comment. The sentence was changed as follows:

 The material was the thermoplastic polymer polypropylene (line 167-168).

  1. The line stating: "the material was considered to be from Exxon Chemical", was it considered or purchased from that company already?

Answer: Thanks for the observation. We considered the material features from Exxon Chemical material to simulate the optimization models. We updated the sentences as follows (lines 168-169): 

The mechanical features of the material were considered by Exxon Chemical (United States) for the simulation.

  1. The conclusion needs to be modified to summarize the research outcomes in short statements with clear observations.

Answer.- Thanks for the punctual observation. The Conclusion section was updated to summarize the research outcomes.